# Research

Subject Areas:
psychology/behaviour

Keywords:
cooperation, experiment, human behaviour, Prisoner's Dilemma

Author for correspondence:
Andreas Orland
e-mail: aorland@uni-potsdam.de

# Cooperation in the Prisoner's Dilemma: an experimental comparison between pure and mixed strategies

Leonie Heuer[1] and Andreas Orland[2]

[1]Department of Economics, FU Berlin, Garystrasse 21, 14195 Berlin, Germany
[2]Department of Economics, University of Potsdam, August-Bebel-Strasse 89, 14482 Potsdam, Germany

 AO, 0000-0001-6954-3669

Cooperation is—despite not being predicted by game theory—a widely documented aspect of human behaviour in Prisoner's Dilemma (PD) situations. This article presents a comparison between subjects restricted to playing pure strategies and subjects allowed to play mixed strategies in a one-shot symmetric PD laboratory experiment. Subjects interact with 10 other subjects and take their decisions all at once. Because subjects in the mixed-strategy treatment group are allowed to condition their level of cooperation more precisely on their beliefs about their counterparts' level of cooperation, we predicted the cooperation rate in the mixed-strategy treatment group to be higher than in the pure-strategy control group. The results of our experiment reject our prediction: even after controlling for beliefs about the other subjects' level of cooperation, we find that cooperation in the mixed-strategy group is lower than in the pure-strategy group. We also find, however, that subjects in the mixed-strategy group condition their cooperative behaviour more closely on their beliefs than in the pure-strategy group. In the mixed-strategy group, most subjects choose intermediate levels of cooperation.

## 1. Introduction

Prisoner's Dilemma (PD) is a social dilemma in which (usually) two players simultaneously face a choice between two options: to cooperate or to defect. The game matrix of the PD with payoffs $T > R > P > S$ is displayed in table 1 (the first payoff in each cell belongs to Player A, the second to Player B). If both players cooperate, they both receive payoff $R$ (for reward). If both players defect, they receive $P$ (for punishment). If only one player defects and the other player cooperates, the defector

**Table 1.** The general PD in matrix form.

| | | Player B | |
|---|---|---|---|
| | | cooperate | defect |
| Player A | cooperate | R, R | S, T |
| | defect | T, S | P, P |

receives $T$ (for temptation) and the cooperator gets $S$ (for sucker). In one-shot interactions (or when the game is repeated for a finite number of periods), each player, independent of the other player's choice, has an incentive to defect. Each player is tempted to maximize his or her own gains by defecting, but if both defect, both lose compared to the situation in which both cooperate. According to the Nash Equilibrium, in equilibrium, no player has an incentive to unilaterally deviate from his or her choice. However, the PD's Nash Equilibrium of mutual defection is not socially efficient. Therefore, it is in society's interest to know the determinants of cooperation in the PD, because this dilemma is likely to arise anywhere conflicts of interest exist—whether in politics [1], economics [2] or even evolution [3]. Due to the frequent occurrence and importance of the PD, many scientific disciplines analysed cooperation in the PD, e.g. evolutionary biology/genetics [4], chaos theory [5], sociology [6], psychology [7] and (experimental) economics [8–13]. As an exhaustive literature review is beyond the scope of this paper, we refer to survey articles for an overview [14,15].

This article presents a comparison between subjects restricted to playing pure strategies and subjects allowed to play mixed strategies in a one-shot symmetric PD laboratory experiment. Our research is motivated by different interpretations of the mixed Nash Equilibrium in the game theoretical literature in economics. Von Neumann & Morgenstern [16, p. 144] interpret mixed strategies as a deliberate randomizing device to conceal one's intentions (e.g. a player in the Matching Pennies game who wants to outsmart his or her opponent) whereas Rosenthal [17, pp. 1353–1354] interprets them as the distribution of pure choices in a large population (e.g. there are two subpopulations, each has its own set of pure strategies available, and nature matches players from the different populations randomly). Yet another interpretation is provided by Aumann [18, p. 3, pp. 15–17] who states that the probabilistic nature of the mixed strategies reflects the uncertainty of the players who do not know what actions the other players take. According to this interpretation, each player always chooses a definite pure strategy, with no attempt to randomize. (Rubinstein [19, pp. 912–915] provides a discussion of these different interpretations.) With our paper, we test whether the rate of cooperation differs when we compare players who are allowed to play mixed strategies (resembling the interpretation in [16]) with players who individually have to choose among their pure strategies (resembling [17]). In order to account for the role of uncertainty about others' actions in [18], we also elicit beliefs about these actions. We chose the PD as the underlying game in our experiment (i) because of its prominent role in various disciplines, and (ii) because cooperation in the PD involves subjects trading off individual and societal interest. We consider cooperation in the PD a more interesting (and moral, see [20]) question than merely one of calculation (like in the Matching Pennies game).

The subjects in one group of our experiment are restricted to playing pure strategies and the subjects in another group are allowed to play mixed strategies in a one-shot symmetric PD game. In both groups, subjects anonymously interact with 10 other subjects and take their decisions all at once (to the best of our knowledge, we are the first to take this approach in a PD experiment). Mixed strategies spare decision-makers from committing themselves to either full cooperation or full defection. Instead, players can select a mix of those pure strategies. The purpose of this study is to determine whether and how the option to play mixed strategies affects cooperation in a PD. In our monetarily incentivized experiment, subjects are divided into two groups which differ only in their decision space. In the control group *Pure*, subjects take a pure-strategy decision. For 10 randomly matched anonymous interactions within their experimental session, they decide all at once to either cooperate or defect according to the game matrix in table 2. In the treatment group *Mixed*, subjects have the option to take a mixed-strategy choice. They decide in *how many* of the 10 randomly matched anonymous interactions they want to cooperate; in the remaining (also randomly matched) anonymous interactions, they defect. Hence, subjects in *Mixed* still have the option to fully cooperate or fully defect, as in *Pure*. The order in which subjects play the chosen mixture of cooperation and defection is randomly determined.

**Table 2.** The game matrix in the experiment (with payoffs in euro cents).

| | | decision of the other subject | |
|---|---|---|---|
| | | cooperate | defect |
| your decision | cooperate | 75 cents, 75 cents | 25 cents, 85 cents |
| | defect | 85 cents, 25 cents | 30 cents, 30 cents |

To increase the chances that decisions are taken deliberately, subjects in both treatments are asked to state a belief about their opponents' cooperativeness. Eliciting these beliefs also allows us to examine the relationship between beliefs and cooperative behaviour more closely. The elicitation of beliefs about other players' behaviour and the consequences of these beliefs for one's own behaviour was an early research topic. Subjects in PD experiments guess that others will play as they themselves intend to play [21–23]. Croson [24] found that when subjects were asked for their best (binary) estimation of what their counterpart in the experiment would do, it decreased subsequent cooperation in one-shot PD experiments by about 30% compared with subjects who were not asked. Acevedo & Krueger [25] attribute this relationship between beliefs and behaviour to evidential reasoning and social value orientation. Rubinstein & Salant [26] present related evidence for self-similarity in strategic interactions akin to the PD.

In a post-experimental questionnaire, we asked subjects about control variables we considered important for experiments conducted with students at a university campus (we decided not to include more control variables as the subjects in the experiments were exclusively students and hence of similar age and educational level and neither of them has participated previously in a PD experiment in the laboratory; we did not include a measure for risk aversion because there is evidence that it does not correlate with behaviour in the PD [27] or the Trust Game [28]). First, we included gender because [29] found females to be more cooperative in the first rounds of a repeated PD experiment (this difference was more pronounced in mixed-gender sessions than when single-gender sessions were compared). See [30, pp. 461–463] for a more general discussion of gender differences in PD experiments and [31] for a meta-study of gender differences in Dictator Game and PD experiments. Second, we included whether subjects had already heard about the experiment (because having heard of the experiment from peers may make subjects behave differently than subjects who have not). Third, we included whether they were familiar with game theory (as the PD is usually taught in game theory classes and knowing the solution may make students behave more in line with theory; see, e.g. [32,33] on the role of subjects' experience in PD experiments). Finally, we asked how many other subjects in the room the subjects knew personally (knowing more of the other subjects personally may make subjects behave more pro-socially, i.e. more prone to cooperate in the PD).

Standard game theory predicts that the option to play mixed strategies in a one-shot PD game will not affect cooperation at all. Mutual defection is the game's only Nash Equilibrium, which means that players have no incentive to unilaterally deviate from the probability distribution of 100% defection and 0% cooperation. Empirically, however, up to 80% of the choices in experimental PD games are cooperative, depending on the calibration of the payoffs [34]. For our experiments, we chose the game matrix presented in table 2. It had already been used in [24], who reported a cooperation rate of 55% and a belief rate of 45%. In *Pure*, pro-social subjects have to face an 'all-or-nothing' decision. Here, uncertainty about others' behaviour is likely to draw pro-social subjects toward defection, because of the fear of being taken advantage of overwhelms the desire to maximize joint outcomes. In *Mixed*, we expect the option to play mixed strategies to encourage pro-social subjects to reciprocally cooperate at least to the same degree that they expect their opponents to cooperate. The crucial point is that only mixing strategies enables subjects to give the best response to their belief. As we expect a distribution very close to 50% cooperation/50% defection of both beliefs and behaviour, the chosen game matrix should give us clear results.

*Prediction:* The cooperation rate in *Mixed* is higher than in *Pure*.

The one-shot decision provides the cleanest test for social dilemmas like the PD. When a decision is only taken once, subjects cannot learn over the course of time (as some subjects gain understanding when feedback is provided [8]). Conditioning one's own behaviour on the observed past behaviour of others is not possible (like the reciprocity reported in Public Goods Game experiments, i.e. in [35]) and reputation-building does not play a role (as it does in [36] when interacting more than once with the same subject).

## 2. Methods

The replication crisis [37–39] has revealed that many results in psychology, experimental economics and other social sciences are not reproducible. We address this crisis by determining the number of required observations with the help of a power calculation (where the expected effect size is based on the literature) before conducting our experiments. Using G*Power 3.1.9.2 [40], a required sample size of 40 in each of the two treatment groups was calculated to provide a statistical power of $1 - \beta = 0.8$ to detect an effect of $d = 0.58$, assuming a one-sided Wilcoxon rank-sum test and an error probability of $\alpha = 0.05$. We used the results in [24] and calculated the effect size based on an expected increase in cooperation of 7 percentage points in *Mixed* over the reported cooperation rate of 55% whose payoff matrix we also use [24, Table 4(a), p. 310]. We assumed a standard deviation of s.d. = 12.16 in both treatments (calculated from the data points in a recent meta-study [13, fig. 3, p. 71]).

A total of 97 students from the University of Potsdam who had subscribed to the ORSEE database (based on [41]) of the Potsdam Laboratory for Economic Experiments (or PLEx, https://www.uni-potsdam.de/plex) were recruited to participate in this experiment. These subjects were randomly assigned to two treatments: 48 subjects in *Pure* and 49 subjects in *Mixed*. A total of 12–18 subjects participated in each of the six sessions conducted in June 2018. Each subject participated in one session only.

After entering the laboratory, subjects were randomly assigned to a computer terminal, after which point any communication between subjects was forbidden. Blinds between workstations prohibited subjects from looking at their neighbours' screens and observing their decisions. A blank sheet of paper and a pen were provided for each subject. Experimental instructions were displayed on the computer screen at the beginning of the experiment (for translations of the experiment and screenshots in German, refer to the repository in the Data Accessibility statement). Sessions were either *Pure* or *Mixed* sessions so that the instructions were identical for all subjects in the room. Each experimental session lasted about 15 min. Subsequent to the experimental game, subjects were asked to fill in a short questionnaire collecting information about subjects' gender (dummy variable `Female` = 1 if female) and whether they had already heard about the experiment (dummy variable `Known Experiment` = 1 if yes), whether they were familiar with game theory (dummy-variable `Game Theory` = 1 if yes) and how many other subjects in the room they knew personally (variable `Known Subjects` = number of known subjects). Subjects earned a show-up fee of €4 and an average of €6.18 in the game (€6.47 in *Pure*, €5.90 in *Mixed*). Subjects received their payoff in private. The experiment was programmed in z-Tree [42] and framed in a neutral way. In both groups, subjects were presented with the payoff matrix in table 2. Cooperation was labelled decision A, defection decision B.

In both groups, subjects had to take one single payoff-relevant decision. In *Pure*, subjects had to decide to play either decision A or decision B in all 10 subsequent interactions (variable `Cooperation`: either 0 or 1, transformed into rates of either 0 or 100). In *Mixed*, in contrast, subjects had to decide in how many of the 10 interactions they would take decision A (variable `Cooperation`: integers between 0 and 10, transformed into rates between 0 and 100). In the remaining interactions, they played decision B. The order in which they played the chosen mix of A or B against their counterparts was randomly determined by the computer. Following this, the computer matched subjects randomly into pairs with one of 10 other subjects in the room. Each subject's payoff from the experiment was the sum of profits earned in the 10 interactions. Subjects did not receive any information about their counterparts or other subjects' decisions.

Before subjects took their decision, they were asked to (non-incentivized) evaluate the other subjects' behaviour. In *Pure*, subjects had to state how many of their 10 interaction partners they expected would choose decision A (variable `Belief`: integer between 0 and 10, transformed into rates between 0 and 100). In *Mixed*, subjects had to state in how many interactions they believed their 10 interaction partners would choose decision A on average (variable `Belief`: number with up to two decimal places between 0 and 10, also transformed into rates).

## 3. Results

### 3.1. Comparison of treatment means

Most important are the comparisons of the means of the two variables of interest, `Cooperation` and `Belief`, in our treatment groups (both variables are expressed here as rates and range between 0 and 100%). We also check our control variables for balanced samples, as differences between treatments may affect the outcomes. Table 3 presents the sample means, differences between treatments and test

**Table 3.** Variable means in both treatments in test of differences. Note: Standard deviations in parentheses and asterisks indicate difference between the treatments.

| variable | Mixed | Pure | difference | test of differences |
|---|---|---|---|---|
| `Cooperation` | 60.000 | 75.000 | $-15.000^{***}$ | two-sided rank-sum, $z = 3.591$, $p = 0.0003$ |
| | (26.141) | (43.760) | | |
| `Belief` | 58.888 | 65.833 | $-6.945^{**}$ | two-sided rank-sum, $z = 2.058$, $p = 0.0396$ |
| | (18.916) | (20.919) | | |
| `Female` | 0.592 | 0.583 | 0.009 | Pearson's $\chi^2$-test, $\chi^2_1 = 0.0072$, $p = 0.932$ |
| | (0.497) | (0.498) | | |
| `Known Subjects` | 0.347 | 0.646 | $-0.299^{**}$ | two-sided rank-sum, $z = 2.066$, $p = 0.0388$ |
| | (0.663) | (0.812) | | |
| `Known Experiment` | 0.061 | 0.167 | $-0.106$ | Pearson's $\chi^2$-test, $\chi^2_1 = 2.6813$, $p = 0.102$ |
| | (0.242) | (0.377) | | |
| `Game Theory` | 0.184 | 0.271 | $-0.087$ | Pearson's $\chi^2$-test, $\chi^2_1 = 1.0504$, $p = 0.305$ |
| | (0.391) | (0.449) | | |
| # observations | 49 | 48 | | |

$^{***}p < 0.01$.

$^{**}p < 0.05$

results on the differences between the treatments. We randomly assign 49 subjects to *Mixed* and 48 subjects to *Pure*. We do not exclude any observations.

In order to compare the (quasi-)continuous variables in the two independent samples, we use the Wilcoxon rank-sum test. It is a non-parametric test as it (in contrast to the *t*-test) neither requires the assumption that both samples are of equal variance nor that the two samples are normally distributed. We apply the $\chi^2$-test to detect differences in the frequencies of binary categories in the two independent samples.

### 3.1.1. Result

Our main question is the difference in cooperation rates between the *Mixed* treatment group and the *Pure* control group. The cooperation rate in *Mixed* is 60%, in *Pure* 75%. A two-sided Wilcoxon rank-sum test shows the difference between the two groups to be highly statistically significant ($p = 0.0003$).

Our prediction that the possibility to play mixed strategies will increase cooperation in the PD is shown to be wrong: the cooperation rate in *Pure* is higher than in *Mixed*.

Beliefs about other subjects' cooperativeness may, of course, also be affected by the decision environment (`Belief` is an endogenous variable). A two-sided Wilcoxon rank-sum test finds the difference between *Mixed* and *Pure* to be statistically different ($p = 0.0396$). Hence, the subjects' beliefs correctly reflect the lower cooperation rate in *Mixed* compared to *Pure*.

In our check for balanced samples, only the variable `Known Subjects` was found to be statistically different between the treatments ($p = 0.0388$). We will later include this variable in a robustness check of the different cooperation rates identified in the two treatments.

## 3.2. Test for gender differences in beliefs and cooperation

Given the interest in gender differences in cooperation mentioned in the introduction, we shortly examine the relationship between gender and cooperation rate and gender and belief separately. We neither observe a statistically significant relationship between `Female` and `Belief` (the Pearson correlation coefficient of both variables is $-0.098$ ($p = 0.5093$) in *Pure* and 0.161 ($p = 0.2706$) in *Mixed*) nor between `Female` and `Cooperation` (the Pearson correlation coefficient of both variables is $-0.068$ ($p = 0.646$) in *Pure* and $-0.020$ ($p = 0.8900$) in *Mixed*).

## 3.3. The relationship between cooperation and beliefs

First, we consider the distributions of the variables `Belief` and `Cooperation`. Figure 1 displays histograms of these two variables in *Pure* and *Mixed*. We observe that the distributions of `Belief` in

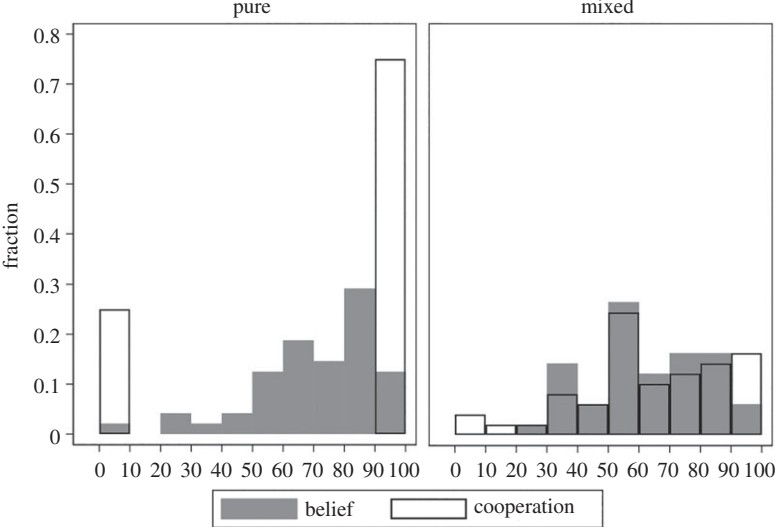

**Figure 1.** Histograms of `Cooperation` and `Belief` by treatment.

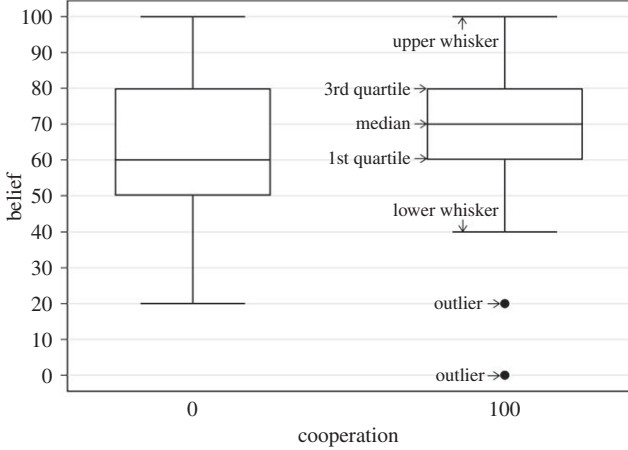

**Figure 2.** Boxplot of `Belief` by `Cooperation` in *Pure*.

the two treatments are similar, with many observations falling in the middle of the interval. Interestingly, in *Mixed*, many subjects also chose intermediate cooperation levels. In this treatment, the distributions of `Belief` and `Cooperation` lie almost on top of one another.

This leads us to the main issue in this section: the relationship between the subjects' beliefs regarding the cooperative play of others and their own decision. Figure 2 shows a boxplot of `Belief` by `Cooperation` in *Pure*. Cooperators have a slightly higher median `Belief` than defectors and their beliefs are more compressed. However, the Pearson correlation coefficient of 0.140 is not found to be significantly different from zero ($p = 0.3445$). Figure 3 shows a scatterplot which suggests a linear relationship between `Cooperation` and `Belief` in *Mixed*. A positive correlation between `Cooperation` and `Belief` in this treatment is confirmed by a Pearson correlation coefficient of 0.403, significantly different from zero ($p = 0.0041$).

## 3.4. Controlling for confounds using OLS regressions

Does the result that subjects in *Mixed* cooperate less than the subjects in *Pure* still hold if we control for the two variables that differed between treatments? Table 4 displays the results from OLS regressions (in economics, the multivariate ordinary least-squares regression is the most common technique to estimate relationships between variables while controlling for covariates' influence). In Model 1, we regress `Cooperation`, using our pooled data, on a constant and a treatment-dummy for *Mixed*. The result confirms our previous finding: significantly more cooperation in *Pure* (*t*-test, $p = 0.043$). In Model 2, we extend Model 1 by adding `Belief` into the regression. Both variables are statistically

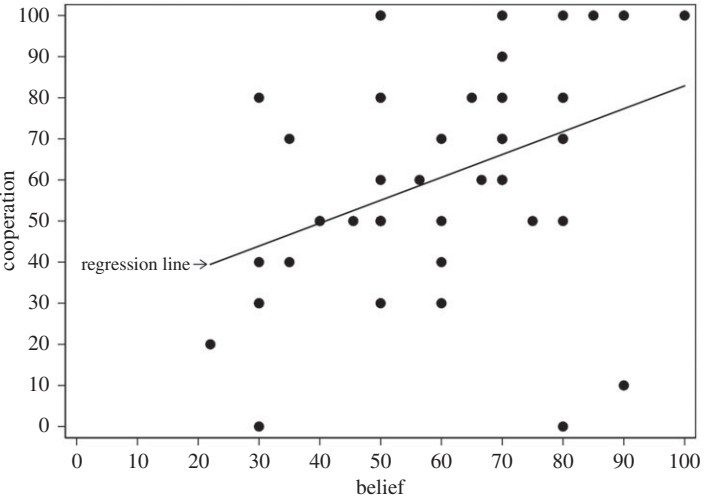

**Figure 3.** Scatterplot of Cooperation by Belief with regression line in *Mixed*.

**Table 4.** Determinants of Cooperation. Note: Standard errors in parentheses and asterisks indicate *t*-test on difference from zero.

|  | Model 1 | Model 2 | Model 3 |
|---|---|---|---|
| constant | 90.000*** | 59.985*** | 58.553*** |
|  | (11.580) | (17.384) | (17.466) |
| *Mixed*-dummy | −15.000** | −12.136* | −11.028 |
|  | (7.301) | (7.256) | (7.359) |
| Belief |  | 0.412** | 0.372** |
|  |  | (0.181) | (0.187) |
| Known Subjects |  |  | 4.652 |
|  |  |  | (5.021) |
| adjusted $R^2$ | 0.033 | 0.073 | 0.072 |
| AIC | 974.4199 | 968.9956 | 970.1043 |
| BIC | 976.9946 | 976.7197 | 980.4031 |
| # observations | 97 | 97 | 97 |

*$p < 0.10$, **$p < 0.05$, ***$p < 0.01$.

different from zero (*t*-tests, *Mixed*-dummy: $p = 0.098$; Belief: $p = 0.025$). Finally, in Model 3, we add Known Subjects to Model 2. Here, the treatment-dummy and Known Subjects are not statistically different from zero (*t*-tests, *Mixed*-dummy: $p = 0.137$; Known Subjects: $p = 0.357$). The subjects' beliefs are significantly different from zero (*t*-test, $p = 0.049$).

Which model provides the best statistical fit (as we neither want to overfit nor underfit our model)? The measure of explained variance, adjusted $R^2$, is highest for Model 2, and the Akaike and Bayesian information criteria (AIC and BIC; the most common criteria for model selection) are lowest for Model 2. All three metrics indicate that Model 2 provides the best statistical fit of the three models. We conclude from this robustness check that the cooperation rate in *Pure* is higher than in *Mixed* even when we control for the variable Belief (which is endogenous to the two treatment groups), contrary to our prediction.

## 4. Conclusion

To summarize, we conducted one-shot PD game experiments. Our treatment variable was the opportunity to play mixed strategies. In a control group, subjects were limited to playing either full

cooperation or full defection against 10 other subjects. In the treatment group, the subjects were allowed to choose any mix of the two strategies. Before subjects took their decision, we elicited their beliefs about the other subjects' level of cooperativeness.

Using a two-sided test, we found that—contrary to our prediction—the cooperation rate in *Pure* was actually higher than in *Mixed*. Even after controlling for the subjects' beliefs in OLS regressions, this difference remains significantly different from zero (though only at the 10% level). As we conducted only a power calculation for a comparison of the treatment averages for Cooperation, we are careful with the interpretation of the higher cooperation rate we detected in *Pure*. However, we see our findings as an indication that cooperation rates differ when subjects can use mixed strategies.

A reviewer of this paper pointed out that the subjects in *Mixed* might cooperate with a certain probability. In *Pure*, these subjects would only cooperate if this probability is higher than a certain threshold (it is very likely that they only cooperate if they believe that more than 50% of the other subjects also cooperate). This switching-point theory sounds reasonable. However, in order to test it one would require an experimental design where each subject goes through both a *Pure* and a *Mixed* stage (the order of *Pure* and *Mixed* should be randomized between subjects and controlled for in the analyses). With our design, we only test whether cooperation rates (and beliefs) differ when subjects can play mixed strategies (between subjects). The beliefs about the cooperativeness of others is endogenous in the control and in the treatment group.

A previous study showed how cooperation rates vary across one-shot symmetric PD experiments when the cooperation/cooperation payoff in the underlying game matrices is varied [43]. They find, as predicted, that the cooperation rate increases when they increase the payoff. They also find that the beliefs about other subjects' behaviour (which were elicited after the subjects took their decision) closely track the cooperation rate in the respective treatment.

We think it would be interesting to combine the experimental design in [43] with our approach. Depending on the parametrization of the PD game matrix, the effect of mixed strategies may be different. When the cooperation rate in a *Pure* treatment is very low, this rate may be higher in a *Mixed* treatment (due to subjects who do not completely defect but choose an intermediate level of cooperation). This, of course, requires another series of experiments. These experiments could also include a questionnaire asking for the subjects' social value orientation in order to disentangle the subjects' motives for cooperating (see [44] for a meta-study of social value orientation in social dilemmas).

Ethics. Economic experiments like ours are not subject to approval by the university's ethical review board (https://www.uni-potsdam.de/senat/kommissionen-des-senats/ek.html). A general informed consent/data privacy statement was signed by all subjects prior to the first experiment at the PLEx. No minors participated in the experiments.

Data accessibility. Data, code, experimental instructions and screenshots are accessible at https://osf.io/p4dgz/.

Authors' contributions. L.H. and A.O. designed the research, conducted the experiments, analysed the results and wrote the manuscript; A.O. programmed the experimental software.

Competing interests. We declare we have no competing interests

Funding. We acknowledge the support of Deutsche Forschungsgemeinschaft (German Research Foundation) and Open Access Publication Fund of University of Potsdam.

Acknowledgements. We are thankful to Lisa Bruttel for her valuable comments. Luis Koch, Fenja Meinecke and Juri Nithammer provided excellent research assistance. Anne Popiel provided proofreading services.

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
