## [Reviewer comments · Royal Society Open Science]

Review History

RSOS-182142.R0 (Original submission)

Review form: Reviewer 1

Is the manuscript scientifically sound in its present form?

No

Are the interpretations and conclusions justified by the results?

No

Is the language acceptable?

Yes

Is it clear how to access all supporting data?

Yes

Do you have any ethical concerns with this paper?

No

Have you any concerns about statistical analyses in this paper?

Yes

Recommendation?

Major revision is needed (please make suggestions in comments)

Comments to the Author(s)

This paper explores whether the possibility of using mixed strategies in one-shot Prisoner's dilemmas favours cooperation. I like the experimental design, which I find to be smart. However, I have major concerns especially about the power analysis, the analysis of the main results (first subsection of the Results section), and the sample size. Therefore, since the analysis of the main result is probably wrong (see below), it is difficult to me to fully judge this paper. Therefore, I think that it currently lies somewhat in the cusp between Major Revision and Rejection. I list all my comments below.

Comments

Introduction

- The PD is not well explained. Please spell out all the strategies and payoffs.
- "According to the Nash Equilibrium, each player's strategy is a best response to the other player's strategy" This sentence is not correct English
- It would be nice to add a general discussion about cooperation. Why is it important? What studies have been done before? Let's not forget that RSOS is a general interest journal with a very broad readership. A set of useful general references for this part could be: Perc & Szolnoki (2010), Perc et al (2013), Rand & Nowak (2013), Capraro (2013), Perc et al (2017), Capraro & Perc (2018)
- Add a motivation: Why are you interested in this particular research question?
- Gender differences in cooperation. You did not refer to the recent meta-analysis by David Rand (Rand, 2017)
- You mention that experience in PD can affect behavior, but you did not include any reference for this non-trivial result. Two useful references are: Rand et al (2014) and Capraro & Cococcioni (2015)
- You say "Empirically, about half the choices in experimental PD games are cooperative". This is simply not true. As it was noted in Capraro et al (2014), the proportion of cooperation highly depends on the cost/benefit ratio, and can be as high as 80% - and probably more, with suitable cost/benefit ratios.
- Since you say that your effect should be driven by people who are afraid of being taken advantage of, perhaps it would have important to collect also a measure of risk aversion? Perhaps worth adding in the discussion?

Methods

- The power analysis is quite weird. $d=0.58$ is a huge effect size, and I'm not sure that corresponds to only 7 percentage points. Moreover, the Results section shows that you found an effect size of 15 per cent, twice as much, but you say that it is not significant. If the power analysis had been correct, then it would have been significant. Something really wrong is going on here.

Results

Comparison of means

- you report both the one-side Wilcoxon test and the two-sides. Usually, the two-side test has a p-value that is twice as much as the p-value of the one-side test. How is it that you do not get this relationship? Are you sure you are reporting the correct analysis?

- The same problem appears in the case of beliefs
- Also, it is not clear to me why you are using both (one-side and two-sides) tests? In your case, you have no a priori reason to assume that one mean is greater than the other one, so I guess you'll need to use the two-side test.

The relationship between cooperation and beliefs

- Not clear what the take home message is, here. To me it seems that, simply, what is happening is that people that, in the mixed condition, would play cooperation with a probability larger than the probability with which they'd play defection, then, in the pure condition, plays cooperation. In other words, it seems to me that the best explanation for these results is that people have a mixed strategy of the form $(aC, (1-a)D)$ in mind. If $a > 1/2$, then, in the Pure condition, they play C. I actually think that this interpretation should be discussed, by the way, since it seems to me that it fits the results quite well. Am I wrong?

Conclusion

- In the Introduction, you mentioned the possibility of gender differences. But I have not seen the analysis of potential gender differences. I think this should be added and discussed in light of the current debate on whether there are gender differences in pro-social behavior. I have already mentioned the meta-analysis by Dave Rand on gender differences in cooperation. Other two useful meta-analyses, one regarding gender differences in altruism (Branas-Garza et al, 2018), the other one regarding gender differences in honesty (Capraro, 2018), might be useful to extend this part of the discussion.
- The sample size is very small and well below current standards, especially since the explosion of the Replicability Crisis. This major limitation should be at the very least acknowledged. Ideally, you might want to consider doing another study, with a larger sample, perhaps on Mechanical Turk. This would make the paper much stronger!

References

- Brañas-Garza P, Capraro V, Rascón-Ramírez E (2018) Gender differences in altruism on Mechanical Turk: Expectations and actual behavior. *Economics Letters* 170, 19-23.
- V Capraro (2013) A model of human cooperation in social dilemmas. *PLoS One* 8 (8), e72427
- Capraro V (2018) Gender differences in lying in sender-receiver games: A meta-analysis. *Judgment and Decision Making* 13, 345-355.
- V Capraro, G Cococcioni (2015) Social setting, intuition, and experience in laboratory experiments interact to shape cooperative decision-making. *Proceedings of the Royal Society B* 282, 20150237
- V Capraro, JJ Jordan, DG Rand (2014) Heuristics guide the implementation of social preferences in one-shot Prisoner's Dilemma experiments. *Scientific Reports* 4, 6790
- V Capraro, M Perc (2018) Grand challenges in social physics: In pursuit of moral behavior. *Frontiers in Physics* 6, 107
- M Perc, J Gómez-Gardeñes, A Szolnoki, LM Floría, Y Moreno (2013) Evolutionary dynamics of group interactions on structured populations: A review. *Journal of the Royal Society Interface* 10, 20120997
- M Perc, JJ Jordan, DG Rand, Z Wang, S Boccaletti, A Szolnoki (2017) Statistical physics of human cooperation. *Physics Reports* 687, 1-51
- M Perc, A Szolnoki (2010) Coevolutionary games – A mini review. *BioSystems* 99, 109-125
- DG Rand, MA Nowak (2013) Human cooperation. *Trends in cognitive sciences* 17 (8), 413-425
- David G Rand, Alexander Peysakhovich, Gordon T Kraft-Todd, George E Newman, Owen Wurzbacher, Martin A Nowak, Joshua D Greene (2014) Social heuristics shape intuitive cooperation. *Nature Communications*, 5.
- Rand DG (2017) Social dilemma cooperation (unlike Dictator Game giving) is intuitive for men as well as women. *Journal of Experimental Social Psychology*, 73, 164-168.

Review form: Reviewer 2

Is the manuscript scientifically sound in its present form?

Yes

Are the interpretations and conclusions justified by the results?

No

Is the language acceptable?

Yes

Is it clear how to access all supporting data?

Not Applicable

Do you have any ethical concerns with this paper?

No

Have you any concerns about statistical analyses in this paper?

No

Recommendation?

Major revision is needed (please make suggestions in comments)

Comments to the Author(s)

In this manuscript, the authors conduct one-shot PD game experiments to compare the levels of cooperation and belief between Pure strategy treatment group and Mixed strategy treatment group. A novel finding in their research is that the cooperation rate in Mixed was not higher than in Pure, which is contrary to their prediction. After reading this manuscript, I have found it interesting. However, there are some remaining issues about the manuscript, requiring some answers.

Major issues:

- 1) Regarding experimental control variables, the authors have not considered some important variables, such as age, educational level, major, reputation, etc. For example, a very important variable is whether subjects often participate in some game experiments? The authors should clarify these.
- 2) During the experiment, I wonder whether the participants in the Mixed strategy treatment group can identify the familiar individuals and then choose to cooperate with them.
- 3) I suspect that the number of subjects selected as objects of interaction would have an impact on the outcome. What happens if subjects interact with more than 10 subjects?
- 4) I think that it's better to describe the data analysis methods, such as one-sided Wilcoxon rank-sum test, two-sided Wilcoxon rank-sum test, OLS regressions, t-test, etc., in the methods section for explanation.
- 5) No detailed legends are given in all the charts and figures in this manuscript.

Minor issues:

- 1) In page 2 line 18, "and subjects allowed to playing the mixed..." -> "and subjects allowed to play the mixed..."
- 2) Page 5, line 48, "[20] show how..." -> "[20] showed how..."

Decision letter (RSOS-182142.R0)

01-Apr-2019

Dear Dr Orland,

The editors assigned to your paper ("Cooperation in the Prisoner's Dilemma: An Experimental Comparison between Pure and Mixed Strategies") have now received comments from reviewers. We would like you to revise your paper in accordance with the referee and Associate Editor suggestions which can be found below (not including confidential reports to the Editor). Please note this decision does not guarantee eventual acceptance.

Please submit a copy of your revised paper before 24-Apr-2019. Please note that the revision deadline will expire at 00.00am on this date. If we do not hear from you within this time then it will be assumed that the paper has been withdrawn. In exceptional circumstances, extensions may be possible if agreed with the Editorial Office in advance. We do not allow multiple rounds of revision so we urge you to make every effort to fully address all of the comments at this stage. If deemed necessary by the Editors, your manuscript will be sent back to one or more of the original reviewers for assessment. If the original reviewers are not available, we may invite new reviewers.

- Data accessibility

If you wish to submit your supporting data or code to Dryad (<http://datadryad.org/>), or modify your current submission to dryad, please use the following link:
<http://datadryad.org/submit?journalID=RSOS&manu=RSOS-182142>

- **Competing interests**

- **Authors' contributions**

- **Acknowledgements**

- **Funding statement**

on behalf of Dr Matjaz Perc (Associate Editor) and Professor Essi Viding (Subject Editor)
openscience@royalsociety.org

Comments to Author:

Reviewers' Comments to Author:

Reviewer: 1

Comments to the Author(s)

This paper explores whether the possibility of using mixed strategies in one-shot Prisoner's dilemmas favours cooperation. I like the experimental design, which I find to be smart. However,

I have major concerns especially about the power analysis, the analysis of the main results (first subsection of the Results section), and the sample size. Therefore, since the analysis of the main result is probably wrong (see below), it is difficult to me to fully judge this paper. Therefore, I think that it currently lies somewhat in the cusp between Major Revision and Rejection. I list all my comments below.

Comments

Introduction

- The PD is not well explained. Please spell out all the strategies and payoffs.
- "According to the Nash Equilibrium, each player's strategy is a best response to the other player's strategy" This sentence is not correct English
- It would be nice to add a general discussion about cooperation. Why is it important? What studies have been done before? Let's not forget that RSOS is a general interest journal with a very broad readership. A set of useful general references for this part could be: Perc & Szolnoki (2010), Perc et al (2013), Rand & Nowak (2013), Capraro (2013), Perc et al (2017), Capraro & Perc (2018)
- Add a motivation: Why are you interested in this particular research question?
- Gender differences in cooperation. You did not refer to the recent meta-analysis by David Rand (Rand, 2017)
- You mention that experience in PD can affect behavior, but you did not include any reference for this non-trivial result. Two useful references are: Rand et al (2014) and Capraro & Cococcioni (2015)
- You say "Empirically, about half the choices in experimental PD games are cooperative". This is simply not true. As it was noted in Capraro et al (2014), the proportion of cooperation highly depends on the cost/benefit ratio, and can be as high as 80% - and probably more, with suitable cost/benefit ratios.
- Since you say that your effect should be driven by people who are afraid of being taken advantage of, perhaps it would have important to collect also a measure of risk aversion? Perhaps worth adding in the discussion?

Methods

- The power analysis is quite weird. $d=0.58$ is a huge effect size, and I'm not sure that corresponds to only 7 percentage points. Moreover, the Results section shows that you found an effect size of 15 per cent, twice as much, but you say that it is not significant. If the power analysis had been correct, then it would have been significant. Something really wrong is going on here.

Results

Comparison of means

- you report both the one-side Wilcoxon test and the two-sides. Usually, the two-side test has a p-value that is twice as much as the p-value of the one-side test. How is it that you do not get this relationship? Are you sure you are reporting the correct analysis?
- The same problem appears in the case of beliefs
- Also, it is not clear to me why you are using both (one-side and two-sides) tests? In your case, you have no a priori reason to assume that one mean is greater than the other one, so I guess you'll need to use the two-side test.

The relationship between cooperation and beliefs

- Not clear what the take home message is, here. To me it seems that, simply, what is happening

is that people that, in the mixed condition, would play cooperation with a probability larger than the probability with which they'd play defection, then, in the pure condition, plays cooperation. In other words, it seems to me that the best explanation for these results is that people have a mixed strategy of the form $(aC, (1-a)D)$ in mind. If $a > 1/2$, then, in the Pure condition, they play C. I actually think that this interpretation should be discussed, by the way, since it seems to me that it fits the results quite well. Am I wrong?

Conclusion

- In the Introduction, you mentioned the possibility of gender differences. But I have not seen the analysis of potential gender differences. I think this should be added and discussed in light of the current debate on whether there are gender differences in pro-social behavior. I have already mentioned the meta-analysis by Dave Rand on gender differences in cooperation. Other two useful meta-analyses, one regarding gender differences in altruism (Branas-Garza et al, 2018), the other one regarding gender differences in honesty (Capraro, 2018), might be useful to extend this part of the discussion.

- The sample size is very small and well below current standards, especially since the explosion of the Replicability Crisis. This major limitation should be at the very least acknowledged. Ideally, you might want to consider doing another study, with a larger sample, perhaps on Mechanical Turk. This would make the paper much stronger!

References

- Brañas-Garza P, Capraro V, Rascón-Ramírez E (2018) Gender differences in altruism on Mechanical Turk: Expectations and actual behavior. *Economics Letters* 170, 19-23.
- V Capraro (2013) A model of human cooperation in social dilemmas. *PLoS One* 8 (8), e72427
- Capraro V (2018) Gender differences in lying in sender-receiver games: A meta-analysis. *Judgment and Decision Making* 13, 345-355.
- V Capraro, G Cococcioni (2015) Social setting, intuition, and experience in laboratory experiments interact to shape cooperative decision-making. *Proceedings of the Royal Society B* 282, 20150237
- V Capraro, JJ Jordan, DG Rand (2014) Heuristics guide the implementation of social preferences in one-shot Prisoner's Dilemma experiments. *Scientific Reports* 4, 6790
- V Capraro, M Perc (2018) Grand challenges in social physics: In pursuit of moral behavior. *Frontiers in Physics* 6, 107
- M Perc, J Gómez-Gardeñes, A Szolnoki, LM Floría, Y Moreno (2013) Evolutionary dynamics of group interactions on structured populations: A review. *Journal of the Royal Society Interface* 10, 20120997
- M Perc, JJ Jordan, DG Rand, Z Wang, S Boccaletti, A Szolnoki (2017) Statistical physics of human cooperation. *Physics Reports* 687, 1-51
- M Perc, A Szolnoki (2010) Coevolutionary games - A mini review. *BioSystems* 99, 109-125
- DG Rand, MA Nowak (2013) Human cooperation. *Trends in cognitive sciences* 17 (8), 413-425
- David G Rand, Alexander Peysakhovich, Gordon T Kraft-Todd, George E Newman, Owen Wurzbacher, Martin A Nowak, Joshua D Greene (2014) Social heuristics shape intuitive cooperation. *Nature Communications*, 5.
- Rand DG (2017) Social dilemma cooperation (unlike Dictator Game giving) is intuitive for men as well as women. *Journal of Experimental Social Psychology*, 73, 164-168.

Reviewer: 2

Comments to the Author(s)

In this manuscript, the authors conduct one-shot PD game experiments to compare the levels of cooperation and belief between Pure strategy treatment group and Mixed strategy treatment group. A novel finding in their research is that the cooperation rate in Mixed was not higher than

in Pure, which is contrary to their prediction. After reading this manuscript, I have found it interesting. However, there are some remaining issues about the manuscript, requiring some answers.

Major issues:

- 1) Regarding experimental control variables, the authors have not considered some important variables, such as age, educational level, major, reputation, etc. For example, a very important variable is whether subjects often participate in some game experiments? The authors should clarify these.
- 2) During the experiment, I wonder whether the participants in the Mixed strategy treatment group can identify the familiar individuals and then choose to cooperate with them.
- 3) I suspect that the number of subjects selected as objects of interaction would have an impact on the outcome. What happens if subjects interact with more than 10 subjects?
- 4) I think that it's better to describe the data analysis methods, such as one-sided Wilcoxon rank-sum test, two-sided Wilcoxon rank-sum test, OLS regressions, t-test, etc., in the methods section for explanation.
- 5) No detailed legends are given in all the charts and figures in this manuscript.

Minor issues:

- 1) In page 2 line 18, "and subjects allowed to playing the mixed..." -> "and subjects allowed to play the mixed..."
- 2) Page 5, line 48, "[20] show how..." -> "[20] showed how..."

Author's Response to Decision Letter for (RSOS-182142.R0)

See Appendix A.

RSOS-182142.R1 (Revision)

Review form: Reviewer 1

Is the manuscript scientifically sound in its present form?

Yes

Are the interpretations and conclusions justified by the results?

Yes

Is the language acceptable?

Yes

Is it clear how to access all supporting data?

Yes

Do you have any ethical concerns with this paper?

No

Have you any concerns about statistical analyses in this paper?

No

Recommendation?

Accept with minor revision (please list in comments)

Comments to the Author(s)

The authors' answer to my criticism regarding the sample size is not convincing. I criticised the sample size (50 participants per condition) by saying that it is well below the current standard, especially after the explosion of the Replicability Crisis (Open Science Collaboration, 2015). The authors responded by saying that their sample size is based on Croson (2000). But this paper is well before the Replicability Crisis, a period in which the standards were different: one might argue that these low standards were indeed the cause of the Replicability Crisis.

I suggested the authors to replicate their findings on Mechanical Turk. They refused to do so and took the occasion to attack experiments on Mechanical Turk on the ground of lack of control and small stakes. But the authors forgot to say that one can easily add controls on AMT and get reliable results (many references) and that stakes do not matter in games involving pro-sociality, as long as they are not too high (thousands of dollars, many references).

In any case, I do not want to make acceptance of this paper conditional on running a new experiment, but I still think that the smallness of the sample and the need to replicate the results should be mentioned as a major limitation of the paper.

Review form: Reviewer 2

Is the manuscript scientifically sound in its present form?

Yes

Are the interpretations and conclusions justified by the results?

Yes

Is the language acceptable?

Yes

Is it clear how to access all supporting data?

Yes

Do you have any ethical concerns with this paper?

No

Have you any concerns about statistical analyses in this paper?

No

Recommendation?

Accept as is

Comments to the Author(s)

In the revised manuscript, the authors have addressed my comments accordingly, and I would like to recommend the publication of the work in Royal Society Open Science.

Decision letter (RSOS-182142.R1)

03-Jun-2019

Dear Dr Orland:

On behalf of the Editors, I am pleased to inform you that your Manuscript RSOS-182142.R1 entitled "Cooperation in the Prisoner's Dilemma: An Experimental Comparison between Pure and Mixed Strategies" has been accepted for publication in Royal Society Open Science subject to minor revision in accordance with the referee suggestions. Please find the referees' comments at the end of this email.

The reviewers and Subject Editor have recommended publication, but also suggest some minor revisions to your manuscript. Therefore, I invite you to respond to the comments and revise your manuscript.

- Ethics statement

- Data accessibility

If you wish to submit your supporting data or code to Dryad (<http://datadryad.org/>), or modify your current submission to dryad, please use the following link:
<http://datadryad.org/submit?journalID=RSOS&manu=RSOS-182142.R1>

- Competing interests

- Authors' contributions

- Acknowledgements

- Funding statement

Because the schedule for publication is very tight, it is a condition of publication that you submit the revised version of your manuscript before 12-Jun-2019. Please note that the revision deadline will expire at 00.00am on this date. If you do not think you will be able to meet this date please let me know immediately.

Supplementary files will be published alongside the paper on the journal website and posted on

the online figshare repository (<https://figshare.com>). The heading and legend provided for each supplementary file during the submission process will be used to create the figshare page, so please ensure these are accurate and informative so that your files can be found in searches. Files on figshare will be made available approximately one week before the accompanying article so that the supplementary material can be attributed a unique DOI.

on behalf of Dr Matjaz Perc (Associate Editor) and Essi Viding (Subject Editor)
openscience@royalsociety.org

Reviewer comments to Author:
Reviewer: 1

Comments to the Author(s)

The authors' answer to my criticism regarding the sample size is not convincing. I criticised the sample size (50 participants per condition) by saying that it is well below the current standard, especially after the explosion of the Replicability Crisis (Open Science Collaboration, 2015). The authors responded by saying that their sample size is based on Croson (2000). But this paper is well before the Replicability Crisis, a period in which the standards were different: one might argue that these low standards were indeed the cause of the Replicability Crisis.

I suggested the authors to replicate their findings on Mechanical Turk. They refused to do so and took the occasion to attack experiments on Mechanical Turk on the ground of lack of control and small stakes. But the authors forgot to say that one can easily add controls on AMT and get reliable results (many references) and that stakes do not matter in games involving pro-sociality, as long as they are not too high (thousands of dollars, many references).

In any case, I do not want to make acceptance of this paper conditional on running a new experiment, but I still think that the smallness of the sample and the need to replicate the results should be mentioned as a major limitation of the paper.

Reviewer: 2

Comments to the Author(s)

In the revised manuscript, the authors have addressed my comments accordingly, and I would like to recommend the publication of the work in Royal Society Open Science.

Author's Response to Decision Letter for (RSOS-182142.R1)

See Appendix B.

Decision letter (RSOS-182142.R2)

07-Jun-2019

Dear Dr Orland,

I am pleased to inform you that your manuscript entitled "Cooperation in the Prisoner's Dilemma: An Experimental Comparison between Pure and Mixed Strategies" is now accepted for publication in Royal Society Open Science.

on behalf of Dr Matjaz Perc (Associate Editor) and Essi Viding (Subject Editor)
openscience@royalsociety.org

Appendix A

Response to the referees

Thank you very much for your comments. We tried to answer your points in the paper and reply to your comments below. We think that the paper improved by your comments!

Andreas Orland

(on behalf of the authors)

Reviewer: 1

Introduction

- The PD is not well explained. Please spell out all the strategies and payoffs.
We spelled out the strategies for both players and added an additional table with a general PD (p. 2). We are convinced that understanding the PD is now much easier.
- "According to the Nash Equilibrium, each player's strategy is a best response to the other player's strategy" This sentence is not correct English
We changed the text (p. 2).
- It would be nice to add a general discussion about cooperation. Why is it important? What studies have been done before? Let's not forget that RSOS is a general interest journal with a very broad readership. A set of useful general references for this part could be: Perc & Szolnoki (2010), Perc et al (2013), Rand & Nowak (2013), Capraro (2013), Perc et al (2017), Capraro & Perc (2018)
We tried to incorporate all the references (p. 2).
- Add a motivation: Why are you interested in this particular research question?
Our motivation is based on different interpretations of mixed strategies.
We added a paragraph (p. 2) explaining our motivation.
- Gender differences in cooperation. You did not refer to the recent meta-analysis by David Rand (Rand, 2017)
We incorporated the reference (p. 3).
- You mention that experience in PD can affect behavior, but you did not include any reference for this non-trivial result. Two useful references are: Rand et al (2014) and Capraro & Cococcioni (2015)
We incorporated the references (p. 3).
- You say "Empirically, about half the choices in experimental PD games are cooperative". This is simply not true. As it was noted in Capraro et al (2014), the proportion of cooperation highly depends on the cost/benefit ratio, and can be as high as 80% - and probably more, with suitable cost/benefit ratios.
We changed the text and added the reference (p. 4).

- Since you say that your effect should be driven by people who are afraid of being taken advantage of, perhaps it would have important to collect also a measure of risk aversion? Perhaps worth adding in the discussion?

We did not include a measure for risk aversion in our post-experimental questionnaire as risk (possibility of losing something due to “force of nature”) is not equivalent (or not considered by subjects equivalent in many lab studies) to strategic uncertainty (possibility of losing something due to other peoples’ behaviour). Previous studies did not find a correlation between risk orientation/risk aversion with behavior in the PD or trust game (Dolbear Jr, F. T., & Lave, L. B. (1966). Risk orientation as a predictor in the Prisoner's Dilemma. *Journal of Conflict Resolution*, 10(4), 506-515; Houser, D., Schunk, D., & Winter, J. (2010). Distinguishing trust from risk: An anatomy of the investment game. *Journal of Economic Behavior & Organization*, 74(1-2), 72-81).

We shortly discuss this (p. 3).

Methods

- The power analysis is quite weird. $d=0.58$ is a huge effect size, and I'm not sure that corresponds to only 7 percentage points. Moreover, the Results section shows that you found an effect size of 15 per cent, twice as much, but you say that it is not significant. If the power analysis had been correct, then it would have been significant. Something really wrong is going on here.

We tried to make our text regarding the power calculation clearer (p. 4). For your convenience, we also include a screenshot of the G*Power screen that shows the calculation of the effect size (small window in the lower right screen) and the number of required observations (in the main window).

We hope to rule out your concerns.

Results

Comparison of means

- you report both the one-side Wilcoxon test and the two-sides. Usually, the two-side test has a p-value that is twice as much as the p-value of the one-side test. How is it that you do not get this relationship? Are you sure you are reporting the correct analysis?

We are sure that we report the correct results. You can find the dataset and the Stata .do-file in an OSF repository. For your convenience, we include screenshots of the Stata output here.

```
. ranksum cooperation, by(treatmenttable2) porder
```

Two-sample Wilcoxon rank-sum (Mann-Whitney) test

treatment~2	obs	rank sum	expected
1	49	1927	2401
2	48	2826	2352
combined	97	4753	4753

unadjusted variance 19208.00
adjustment for ties -1781.93

adjusted variance 17426.07

Ho: cooper~n(treatm~2==1) = cooper~n(treatm~2==2)
z = -3.591
Prob > |z| = 0.0003

P{cooper~n(treatm~2==1) > cooper~n(treatm~2==2)} = 0.298

- The same problem appears in the case of beliefs

Here is the Stata output:

```
. ranksum belief, by(treatmenttable2) porder
```

Two-sample Wilcoxon rank-sum (Mann-Whitney) test

treatment~2	obs	rank sum	expected
1	49	2119	2401
2	48	2634	2352
combined	97	4753	4753

unadjusted variance 19208.00
adjustment for ties -431.15

adjusted variance 18776.85

Ho: belief(treatm~2==1) = belief(treatm~2==2)
z = -2.058
Prob > |z| = 0.0396

P{belief(treatm~2==1) > belief(treatm~2==2)} = 0.380

- Also, it is not clear to me why you are using both (one-side and two-sides) tests? In your case, you have no a priori reason to assume that one mean is greater than the other one, so I guess you'll need to use the two-side test.

Thank you. We now only report the two-sided results.

The relationship between cooperation and beliefs

- Not clear what the take home message is, here. To me it seems that, simply, what is happening is that people that, in the mixed condition, would play cooperation with a probability larger than the probability with which they'd play defection, then, in the pure condition, plays cooperation. In other words, it

seems to me that the best explanation for these results is that people have a mixed strategy of the form $(aC, (1-a)D)$ in mind. If $a > 1/2$, then, in the Pure condition, they play C. I actually think that this interpretation should be discussed, by the way, since it seems to me that it fits the results quite well. Am I wrong?

We discuss your theory of switching points in the conclusion (p. 9).

Conclusion

- In the Introduction, you mentioned the possibility of gender differences. But I have not seen the analysis of potential gender differences. I think this should be added and discussed in light of the current debate on whether there are gender differences in pro-social behavior. I have already mentioned the meta-analysis by Dave Rand on gender differences in cooperation. Other two useful meta-analyses, one regarding gender differences in altruism (Branas-Garza et al, 2018), the other one regarding gender differences in honesty (Capraro, 2018), might be useful to extend this part of the discussion.

We examine cooperation in the PD. Branas-Garza et al. (2018) examine the Dictator Game, Capraro (2018) examines the Sender-Receiver Game.

We chose not to include these references. Croson & Gneezy (2009) serve as our main reference to gender differences in economic experiments and preferences.

We included a short section on gender differences (which we cannot find) in the paper. See p.6.

- The sample size is very small and well below current standards, especially since the explosion of the Replicability Crisis. This major limitation should be at the very least acknowledged. Ideally, you might want to consider doing another study, with a larger sample, perhaps on Mechanical Turk. This would make the paper much stronger!

Our sample size is comparable to Croson (2000) and based on a power calculation. We did our best to react to our professions' problems revealed by the replication crisis. Also, we intend not to run more tests and regressions with 97 observations than originally spelled out in the Methods section... We want to compare aggregate cooperation rates and control for covariates that differ between groups (see one of the introductory examples in Angrist & Pischke (2011), Mostly harmless econometrics, Chapter 2, pp. 18-20, we follow a similar approach). MTurk studies come problems on their own. Lack of control is one. In our lab experiments, there was no communication between subjects in our lab; subjects have not previously participated in a PD experiment in the

respective lab (see problems with subjects in experiments who participate often in some experiments; Capraro V. & Cococcioni G. 2015), and we included control variables that account for problems that might arise due to a student sample. In MTurk experiments, subjects are far more heterogeneous in many dimensions. Hence we do not consider many variables that are indeed important like level of education, income, English as first language, etc. Our sample is rather homogeneous in many ways. We see this as an advantage.

Also, in our experiment, subjects on average earned about 10 euros in an experiment that took 15 minutes. The monetary incentive is higher than in many MTurk studies that pay much less (e.g., Amir et al. 2012, Economic Games on the Internet. Plos One; they paid one US-Dollar). Thus, we expect that our design is salient.

Allow us to disagree: We think that we present a clean study that does not pretend to achieve more than we state in the paper.

Reviewer: 2

In this manuscript, the authors conduct one-shot PD game experiments to compare the levels of cooperation and belief between Pure strategy treatment group and Mixed strategy treatment group. A novel finding in their research is that the cooperation rate in Mixed was not higher than in Pure, which is contrary to their prediction. After reading this manuscript, I have found it interesting. However, there are some remaining issues about the manuscript, requiring some answers.

Major issues:

1) Regarding experimental control variables, the authors have not considered some important variables, such as age, educational level, major, reputation, etc. For example, a very important variable is whether subjects often participate in some game experiments? The authors should clarify these.

As our subject pool is very homogeneous, we chose not to include more variables.

We added some lines regarding the choice of our covariates (p. 3).

2) During the experiment, I wonder whether the participants in the Mixed strategy treatment group can identify the familiar individuals and then choose to cooperate with them.

In both experimental treatments, all subjects are randomly assigned and cannot choose their interaction partners.

We made this clearer in the text (p. 3).

3) I suspect that the number of subjects selected as objects of interaction would have an impact on the outcome. What happens if subjects interact with more than 10 subjects?

In both groups, each subject could not interact with more (or less) than ten other subjects. The random assignment of subjects was computerized. We made this clearer in the text (p. 3).

4) I think that it's better to describe the data analysis methods, such as one-sided Wilcoxon rank-sum test, two-sided Wilcoxon rank-sum test, OLS regressions, t-test, etc., in the methods section for explanation.

We included short explanations for the Wilcoxon rank sum test and the Chi2 test (p. 6). Beyond that, we expect that the readership of RSOS has some background in introductory statistics (a short look at some other articles in RSOS reveals that basic statistical methods are usually not explained).

5) No detailed legends are given in all the charts and figures in this manuscript.

We added very detailed legends in Figures 2 and 3. (We simply do not know which elements could be added in Figure 1.)

Minor issues:

1) In page 2 line 18, "and subjects allowed to playing the mixed..." → "and subjects allowed to play the mixed..."

We changed the text.

2) Page 5, line 48, "[20] show how..." → "[20] showed how..."

We changed the text.

Appendix B

Response to the referees

Again, thank you for your comments. We explain below how we incorporated your comments into our paper.

Andreas Orland

(on behalf of the authors)

Reviewer: 1

The authors' answer to my criticism regarding the sample size is not convincing. I criticised the sample size (50 participants per condition) by saying that it is well below the current standard, especially after the explosion of the Replicability Crisis (Open Science Collaboration, 2015). The authors responded by saying that their sample size is based on Croson (2000). But this paper is well before the Replicability Crisis, a period in which the standards were different: one might argue that these low standards were indeed the cause of the Replicability Crisis. I suggested the authors to replicate their findings on Mechanical Turk. They refused to do so and took the occasion to attack experiments on Mechanical Turk on the ground of lack of control and small stakes. But the authors forgot to say that one can easily add controls on AMT and get reliable results (many references) and that stakes do not matter in games involving pro-sociality, as long as they are not too high (thousands of dollars, many references). In any case, I do not want to make acceptance of this paper conditional on running a new experiment, but I still think that the smallness of the sample and the need to replicate the results should be mentioned as a major limitation of the paper.

Allow us to disagree. We responded by stating that “our sample size is comparable to Croson (2000) *and based on a power calculation.*” The expected effect size in our power calculation is based on the literature and we collected slightly more observations than required (as we could not conduct sessions with less than eleven subjects).

We now cite articles that we consider important references as starting points about/into the replication crisis in social sciences (p. 4). We think our approach is valid as we intended to only test *one* hypothesis. Our initial submission tested this hypothesis and also included a robustness check (the OLS regressions).

As a researcher, one is always tempted to make most out of the data collected. One example is the question for tests of gender differences you asked for in your last report (now in Section 3b). If that would have been our objective from the outset, we should have adjusted our alpha for this multiple hypotheses testing... Despite our rather low number of observations, you asked for this test. Biases in publication are found in many aspects of research.

Thank you for your criticism. We are always open to improve our research and learn from mistakes. Our paper is better than before!

Reviewer: 2

In the revised manuscript, the authors have addressed my comments accordingly, and I would like to recommend the publication of the work in Royal Society Open Science.

Thank you very much for your comments.